# Eye Fairness: A Large-Scale 3D Imaging Dataset for Equitable Eye Diseases Screening and Fair Identity Scaling

## Abstract

Fairness or equity in machine learning is profoundly important for societal well-being, but limited public datasets hinder its progress, especially in the area of medicine. It is undeniable that fairness in medicine is one of the most important areas for fairness learning's applications. Currently, no large-scale public medical datasets with 3D imaging data for fairness learning are available, while 3D imaging data in modern clinics are standard tests for disease diagnosis. In addition, existing medical fairness datasets are actually repurposed datasets, and therefore they typically have limited demographic identity attributes with at most three identity attributes of age, gender, and race for fairness modeling. To address this gap, we introduce our Eye Fairness dataset with 30,000 subjects (EyeFairness-30k) covering three major eye diseases including age-related macular degeneration, diabetic retinopathy, and glaucoma affecting 380 million patients globally. Our EyeFairness dataset includes both 2D fundus photos and 3D optical coherence tomography scans with six demographic identity attributes including age, gender, race, ethnicity, preferred language, and marital status. We also propose a fair identity scaling (FIS) approach combining group and individual scaling together to improve model fairness. Our FIS approach is compared with various state-of-the-art fairness learning methods with superior performance in the racial, gender, and ethnicity fairness tasks with 2D and 3D imaging data, which demonstrate the utilities of our EyeFairness dataset for fairness learning. To facilitate fairness comparisons between different models, we propose performance-scaled disparity measures, which can be used to compare model fairness accounting for overall performance levels. The dataset and code are publicly accessible via `https://github.com/anonymous4science/EyeFairness`.

## 1 Introduction

The advancement of machine learning and artificial intelligence heavily relies on task-specific public datasets with applications across natural image classification, image captioning, and medical imaging processing Deng et al. (2009); Krizhevsky et al. (2009); Lin et al. (2014); Krishna et al. (2017); Antol et al. (2015); Johnson et al. (2017); Irvin et al. (2019); Johnson et al. (2019). In recent years, the issue of fairness and equity with machine learning models has gained more and more attention from the machine learning and computer vision research community due to its profound importance to our society Kadambi (2021); Parikh et al. (2019); Mehrabi et al. (2021). It undermines our societal values when machine learning models exhibit bias against certain demographic identity groups. However, publicly available datasets for fairness learning are limited both in terms of quantity and quality.

The limitations are in three aspects. First, the vast majority of prior public datasets for fairness learning are in the areas of criminology, education, and finance Dressel & Farid (2018); Asuncion & Newman (2007); Wightman (1998); Miao (2010); Kuzilek et al. (2017); Ruggles et al. (2015); Zhang et al. (2017) in the form of tabular data that are incompatible with many influential deep-learning models that rely on imaging data, and

there are very limited datasets for fairness learning in the area of medicine Asuncion & Newman (2007); Irvin et al. (2019); Johnson et al. (2019); Tschandl et al. (2018); Groh et al. (2021); Zambrano Chaves et al. (2021); odi; Afshar et al. (2021); Farsiu et al. (2014); Wyman et al. (2013), while it is undeniable that fairness in medicine is one of the most important areas for fairness learning's applications. Second, though there are a number of medical datasets for fairness learning Asuncion & Newman (2007); Irvin et al. (2019); Johnson et al. (2019); Tschandl et al. (2018); Groh et al. (2021); Zambrano Chaves et al. (2021); odi; Afshar et al. (2021); Farsiu et al. (2014); Wyman et al. (2013), these datasets are actually repurposed datasets for fairness learning, and therefore they typically have limited demographic identity attributes. Most of these repurposed medical datasets only contain at most three demographic identity attributes, including age, gender, and race Irvin et al. (2019); Johnson et al. (2019). Lastly, prior medical fairness datasets 3D imaging data are all relatively small with sample sizes up to 550 Afshar et al. (2021); Wyman et al. (2013); Farsiu et al. (2014), while 3D imaging data in modern clinics are standard tests for disease diagnosis. Having 3D imaging data would make the study of model fairness closer to clinical reality.

In this paper, we publish a comprehensive medical dataset termed Eye Fairness including three major eye diseases, which are age-related macular degeneration (AMD), diabetic retinopathy (DR), and glaucoma. AMD, DR, and glaucoma affect 20 million, 10 million, and 3 million patients in the US Rein et al. (2022); Lundeen et al. (2023); Gupta et al. (2016), respectively, and 200 million, 103 million, and 80 million patients worldwide Wong et al. (2014); Teo et al. (2021); Tham et al. (2014), respectively. AMD, DR, and glaucoma cause permanent damage to the human retina and result in irreversible vision loss with currently available clinical treatments. Timely detection of these eye diseases is therefore critical for clinicians to initiate treatments to save the remaining vision. However, vision loss in the early stage is asymptotic due to fellow eye and brain compensation. The asymptotic nature of early vision loss coupled with the lack of convenient and affordable ophthalmic care results in a substantial number of patients with eye diseases being undiagnosed Neely et al. (2017); Kovarik et al. (2016); Shaikh et al. (2014). For instance, half of glaucoma patients are undiagnosed Shaikh et al. (2014). The undiagnosed eye disease issue is even more severe in minority groups. For instance, it has been reported that Blacks have 4.4 times greater odds of having undiagnosed and untreated glaucoma than Whites Shaikh et al. (2014), while the disease burden of glaucoma in Blacks is doubled compared with Whites Rudnicka et al. (2006); Friedman et al. (2006). Automated eye disease detection with deep learning using retinal imaging is promising to provide affordable eye disease screening to alleviate societal disease burden and reduce health disparities between different demographic identity groups, which can be deployed in primary care and pharmacies without needing the subjects to visit the more expensive and busy ophthalmology clinics. However, potential deep learning systems for automated eye disease screening should address potential fairness issues prior to clinical implementation. Numerous studies have been performed to use deep learning for eye disease screening with a number of public datasets Li et al. (2019a); Orlando et al. (2020); Bajwa et al. (2020); Diaz-Pinto et al. (2019); Gulshan et al. (2016); Li et al. (2019b; 2021); odi; apt; Farsiu et al. (2014); Liu et al. (2014); Fang et al. (2022). Yet, all existing public datasets for eye disease screening lack comprehensive demographic identity attributes for fairness learning, which results in the fact that no published studies have been performed to assess and address the fairness issue in eye disease screening.

The highlights of our dataset are as follows: (1) The first large-scale fairness dataset totaling 30,000 samples in medical imaging with comprehensive demographic identity attributes including age, gender, race, ethnicity, preferred language, and marital status. (2) We have access to 3D imaging data of optical coherence tomography scans in addition to 2D imaging data of scanning laser ophthalmoscopy fundus images. This provides the opportunity for 3D fairness learning, which has been largely unexplored in the literature due to the lack of public datasets.

In addition to our valuable dataset, we propose a fair identity scaling method to address the fairness issue in eye disease screening as an additional contribution. Fair identity scaling uses learnable group weights and individual loss information in the previous training batch to scale the loss function in the current training batch. Samples with higher group weights

and individual loss values in the previous training batch would carry higher weights in the loss function in the current training batch. The rationale for combining group and individual scaling is that group scaling alone ignores the within-group sample characteristic variations, which may unnecessarily overweight or underweight most samples in an identity group due to isolated individual outliers leading to model deterioration. Combining group and individual scaling is promising to both address the group fairness issue and tackle the within-group sample variations.

To facilitate fairness comparisons between different models, we propose a performance-scaled disparity measure. The motivation to propose the performance-scaled performance measure is that current fairness metrics such as demographic parity difference (DPD) and difference in equalized odds (DEOdds) Zietlow et al. (2022); Jiang et al. (2021) may not adequately account for the trade-off between accuracy and fairness. In other words, a model with the same fairness could have quite different accuracy performance, which is not reflected by different fairness metrics such as DPD and DEOdds. Our performance-scaled disparity measure can be flexibly used to compare all kinds of fairness metrics in the context of overall performance, such as the area under the receiver operating characteristic curve (AUC).

Our core contributions are summarized as follows: (1) We introduce the first large-scale fairness learning dataset with 30,000 subjects and six demographic identity attributes for eye disease screening including three major eye disorders affecting about 380 million people worldwide; (2) We develop a novel fair identity scaling approach to promote model fairness with both group and individual scaling; (3) We design a new performance-scaled disparity metric to evaluate model performance across different models.

## 2 RELATED WORK

**Medical Fairness Datasets**. It is known that the burden of many common diseases is greater in socioeconomically disadvantaged minority groups. However, minority groups are underdiagnosed due to a lack of access to affordable healthcare. Automated disease detection by deep learning has been recognized as an affordable way to reduce healthcare disparity against minority groups. However, before such a deep learning screening system can be used in practice, it has to be evaluated against potential model performance inequality, which needs to be mitigated, if any. A couple of medical fairness datasets (**Table 1**) have been available to the public for fairness learning including 2D datasets of CheXpert Irvin et al. (2019), MIMIC-CXR Johnson et al. (2019), Fitzpatrick17k Groh et al. (2021), HAM10000 Tschandl et al. (2018) and OL3I Zambrano Chaves et al. (2021) and 3D datasets of COVID-CT-MD Afshar et al. (2021), ADNI 1.5T Wyman et al. (2013) and AMD-OCT Farsiu et al. (2014). While the sample sizes for the 2D medical fairness datasets are large enough (e.g. 222,793 images for CheXpert Irvin et al. (2019) and 370,955 for MIMIC-CXR Johnson et al. (2019)), the sample sizes for 3D medical fairness datasets are only up to 550 images Afshar et al. (2021); Wyman et al. (2013); Farsiu et al. (2014). In this paper, we will release a large-scale medical fairness dataset with 30,000 OCT images with each including 128 or 200 B-scans and 30,000 SLO fundus images that are ready for fairness learning with identity attributes of age, gender, race, ethnicity, preferred language, and marital status. Our dataset covers three major eye diseases consisting of AMD, DR, and glaucoma affecting 380 million patients globally.

**Fairness Models**. Prior fairness learning models mainly take four distinct approaches including fair data representation, fair feature encoding, fair loss constraint, and fair batch training. The fair data representation approaches Ramaswamy et al. (2021); Zhang & Sang (2020); Zietlow et al. (2022) leverage data generation and data augmentation schemes to improve data representation fairness across different identity groups. For example, Ramaswamy and coworkers used generative adversarial networks to generate realistic-looking images and perturb these images in the underlying latent space to generate training data that is balanced for each protected attribute to improve model fairness. The fair feature encoding approaches Zhang et al. (2018); Beutel et al. (2017); Roh et al. (2020) use regularization terms to either enforce the latent features to be predictive or unpredictive of respective demographic attributes. For instance, Sarhan and coworkers proposed to explicitly enforce the meaningful representation to be agnostic to sensitive information by

Table 1: Public medical fairness datasets.

| Dataset | Imaging Modality | Number of Images | Identity Attribute | 3D |
|---|---|---|---|---|
| CheXpert Irvin et al. (2019) | Chest X-ray | 222,793 | Age; Gender; Race | ✗ |
| MIMIC-CXR Johnson et al. (2019) | Chest X-ray | 370,955 | Age; Gender; Race | ✗ |
| Fitzpatrick17k Groh et al. (2021) | Skin photos | 16,012 | Skin type; | ✗ |
| HAM10000 Tschandl et al. (2018) | Dermatoscopy | 9,948 | Age; Gender | ✗ |
| OL3IZambrano Chaves et al. (2021) | Heart CT | 8,139 | Age; Gender | ✗ |
| ODIR-2019 odi | Fundus | 8,000 | Age; Gender | ✗ |
| COVID-CT-MDAfshar et al. (2021) | Lung CT | 308 | Age; Gender | ✓ |
| AMD-OCTFarsiu et al. (2014) | OCT | 384 | Age | ✓ |
| ADNI 1.5T Wyman et al. (2013) | Brain MRI | 550 | Age; Gender | ✓ |
| Eye Fairness | Fundus and OCT | Fundus: 30,000; OCT: 30,000 (each with 128 or 200 B-Scans) | Age; Gender; Race; Ethnicity; Preferred Language; Marital Status | ✓ |

entropy maximization Sarhan et al. (2020). The fair loss constraint approaches adapt the standard loss function with model fairness metrics such as demographic parity difference (DPD) and difference in equalized odds (DEOdds) Hardt et al. (2016); Agarwal et al. (2018; 2019). For instance, Agarwal and coworkers explored improving model fairness by restricting prediction error to any protected group to be below some pre-determined level Agarwal et al. (2019). Fair batch training seeks to update the training loss function iteratively based on the latest group-wise model fitting information Donini et al. (2018); Sagawa et al. (2019). For instance, the group distributionally robust optimization minimizes the maximum training loss across all identity groups with increased regularization across the training steps Sagawa et al. (2019). In this paper, We propose a fair identity scaling method combining group and individual scaling to improve model fairness. The rationale of our fair identity scaling method combining group and individual scaling is that group scaling itself does not account for within-group sample characteristic variations and therefore may unnecessarily overweight or underweight most samples in an identity group due to outliers, which may lead to unfavorable results. Compared with existing fairness learning models typically only using group-level information to address model equity issues, we additionally consider individual sample variations within each identity group. We hypothesize that combining group and individual scaling may outperform existing fairness learning models.

**Fairness Metrics**. There are three prevalent fairness metrics grounded on distinct assumptions, namely, demographic parity difference (DPD) Bickel et al. (1975); Agarwal et al. (2018; 2019), difference in equal opportunity (DEO) Hardt et al. (2016), and difference in equalized odds (DEOdds) Agarwal et al. (2018). Demographic parity Agarwal et al. (2018; 2019) aims to ensure that a predictive model's outputs are uninfluenced by an individual's affiliation with a sensitive group, attaining demographic parity when there exists no linkage between prediction probabilities and such group affiliation, symbolizing uniform selection rates across groups with a DPD of 0. Conversely, DEO Hardt et al. (2016) emphasizes equalizing the true positive rate (TPR) of predictions across groups delineated by a sensitive attribute (e.g., race or gender), realizing equal opportunity when TPR is consistent across groups, implying that positive predictions are made at an identical rate for true positive class members in each group. DEOdds Agarwal et al. (2018) expands on DEO, necessitating prediction impartiality from sensitive group affiliation, wherein groups maintain equal false positive and true positive rates.

A main shortcoming of the existing fairness metrics DPD, DEO, and DEOdds is that their relationship with model performance metrics is unclear, while clinicians are most concerned about fairness in the context of overall performance level. The same level of fairness at different performance levels could mean very different things to clinicians and patients. In medical research, fairness metrics that are more intuitive to be understood by clinicians with clearer links to performance levels are needed. Therefore, in this paper, we propose the performance-scaled disparity (PSD) metrics to measure model fairness. Specifically, the PSD metrics are calculated as the standard deviation of group performance or absolute maximum group performance difference divided by overall performance.

## 3 DATASET ANALYSIS

This study strictly adheres to the principles outlined in the Declaration of Helsinki, and has been approved by our institute's Institutional Review Board. All data in this dataset are de-identified.

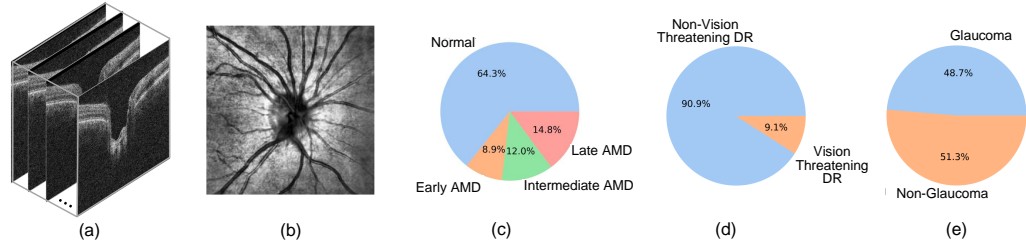

Figure 1: (a) 3D OCT B-scans. (b) SLO fundus image. (c) The label distribution for AMD, (d) The label distribution for DR, and (e) The label distribution for glaucoma.

**Data Composition.** Our dataset including three types of measurements consisting of (1) retinal imaging data, (2) demographic identity group information, and (3) disease diagnosis for three major eye diseases damaging the human retina causing irreversible blindness including AMD, DR, and glaucoma, impacting 380 million patients globally. (1) Retinal imaging data: we have both 2D SLO fundus images measuring the retinal surface and 3D optical coherence tomography measuring the in-depth retinal layer structures, as shown in **Figure 1**. Both the 2D SLO fundus images and 3D OCT scans can effectively assess retinal abnormalities due to eye diseases, while OCT is known to be superior in diagnostic accuracy. (2) Demographic identity group attributes: there are six identity group attributes available in this dataset based on self-reported patient information including age, gender, race, ethnicity, preferred language, and marital status. (3) Disease diagnosis: we have disease diagnosis for AMD, DR, and glaucoma. For AMD and DR, the diagnostic information was extracted from the International Classification of Diseases (ICD) codes in the patient's electronic health records. For Glaucoma, the disease diagnosis is defined based on the patient's visual function. Specifically, the subjects in the AMD dataset are categorized into four classes including normal, early AMD, intermediate AMD, and late AMD (both dry and wet AMD in the late stage), the subjects in the DR dataset are categorized into two classes including non-vision threatening DR and vision-threatening DR Bellemo et al. (2019), and the subjects in the glaucoma dataset are categorized into two classes including normal (visual function measured by visual field [VF] test is normal: VF mean deviation $\geq$ -1 dB and normal VF glaucoma hemifield test and pattern standard deviation results) and glaucoma (visual function measured by VF test is abnormal: VF mean deviation $<$ -3 dB and abnormal VF glaucoma hemifield test and pattern standard deviation results).

**Data Characteristics.** Our dataset includes 10,000 subjects for AMD, DR, and glaucoma separately, totaling 30,000 subjects. The proportions of the four AMD classes (**Figure 1 [c]**) are: normal with 64.3%, early AMD with 8.9%, intermediate AMD with 12.0%, and late AMD with 14.8%. The proportion of vision-threatening DR (**Figure 1 [d]**) is 9.1% compared with 90.9% non-vision-threatening DR. The proportion of glaucoma (**Figure 1 [e]**) is 48.7% compared with 51.3% normal. Note that, as we require all subjects to have VF tests, which bias our sample toward the side of more glaucoma patients. However, using VF test to label subjects is more reliable and consistent compared with clinicians' judgment. The demographic characteristics for 30,000 subjects in the dataset are detailed as follows. The average age is 64.1 $\pm$ 17.0 years. The self-reported patient demographic information is as follows: Gender: female: 57.1% and male: 42.9%; Race: White: 78.6%, Black: 13.7%, and Asian: 7.7%; Ethnicity: Hispanic: 3.8% and non-Hispanic: 96.2%; Preferred language: English: 91.6%, Spanish: 1.8%, other languages: 5.9%, and 4.9%, single: 24.6%, divorced: 7.0%, legally separated: 0.9%, and widowed: 8.6%.

## 4 METHODOLOGY

**Problem Set-Up.** With the labeled data $\mathcal{D} = \{(x_i, y_i, a_i)\}$, where $x \in \mathbb{R}^d$ is an OCT Bscan sample or SLO fundus image, $y \in \mathcal{Y}$ is a disease diagnosis label such as AMD, and $a \in \mathcal{A}$ is an identity attribute associated with the patient, such as gender, race, or ethnicity. In a conventional supervised learning paradigm, the training process aims to find a model $f \in \mathcal{F} : \mathbb{R}^d \xrightarrow{\theta} \mathcal{Y}$ with the parameters $\theta$ to maximize classification accuracy. In contrast, in the fairness learning scheme, we have to take identity information into account when training

a model, *i.e.*, $f \in \mathcal{F} : \mathbb{R}^d \times \mathcal{A} \xrightarrow{\theta} \mathcal{Y}$. Correspondingly, fairness learning should also minimize discrepancies between different identity groups in addition to maximizing accuracy.

**Fair Identity Scaling**. Our fair identity scaling model employs learnable group weights and past individual loss data to adjust the loss function during the current training batch. Essentially, samples that had higher group weights and individual loss values in the prior batch will be given more weight in the current batch's loss function. This approach of combining both group and individual scaling is taken to not only address fairness at a group level but also manage within-group sample variations. This is done to avoid issues that may arise if only group scaling is used, as it could overly weight or underweight most samples within a group due to isolated outliers, consequently deteriorating the model. Mathematically, fair identity scaling is formulated as follows:

$$\mathcal{L}(f, \mathcal{B}^t) = \frac{1}{|\mathcal{B}^t|} \sum_{(x_i, y_i, a_i) \in \mathcal{B}^t} \left[ |\mathcal{B}^t| \frac{\exp((c \cdot \beta_{a_i}^{t-1} + (1-c) \cdot \ell_i^{t-1})/\tau)}{\sum_{j=1}^{|\mathcal{B}^t|} \exp((c \cdot \beta_{a_j}^{t-1} + (1-c) \cdot \ell_j^{t-1})/\tau)} \cdot \ell_i^t \right], \quad (1)$$

where $\ell_i^t$ is cross-entropy loss over a mini-batch $\mathcal{B}^t$ at the training step t. Group scaling is achieved by a learnable weighting parameter $\beta_a$ that is identity group-specific. Individual scaling is realized with the term $\ell_i$ in the softmax function. The fusion weight $c$ ranging between 0 and 1 is used to control group and individual scaling integration, where $c = 1$ means group scaling alone, and $c = 1$ indicates individual scaling alone. $\tau$ is a temperature scaling parameter. When $\tau \to +\infty$, Equation (1) simplifies to the vanilla version of the loss function, denoted as $\mathcal{L}(f, \mathcal{B}^t) = \frac{1}{|\mathcal{B}^t|} \sum_{i=1}^{|\mathcal{B}^t|} \ell_i^t$.

# 5 Experimental Setup & Results

## 5.1 Experimental Setup

**Dataset Split**. For each of the AMD, DR, and glaucoma datasets, we have 10,000 samples from 10,000 subjects. We use 6,000, 1,000, and 3,000 subjects for training, validation, and testing. Random split is used for each model run.

**Method**. We use EfficientNet-B1 Tan & Le (2019) as our backbone model for 2D fundus images, which is widely considered as one of the best backbone models for medical imaging tasks. We use a VGG-derived 3D model for 3D OCT scans, which also has been demonstrated to have robust performance Simonyan & Zisserman (2014). There are numerous fairness learning models available to test. Given the limited paper space, we carefully choose two SOTA models including the fair adversarial training (Adv) Beutel et al. (2017) and the fair contrastive loss Wang et al. (2022) (FSCL) for comparisons, which are recognized as the two most robustly performed fairness learning models. Since the FSCL model is based on image augmentation, we only use it with 2D fundus images as effective image augmentation strategies for 3D imaging data are largely unclear in the literature. We adhere to their official code repositories' standard experimental protocol and hyper-parameter choices.

**Metrics**. To facilitate the model fairness assessment, we use both overall and group-wise AUCs to compare model performance. In addition, we will use traditional fairness metrics of DPD and DEOdds to assess model fairness. Furthermore, we propose to use our performance-scaled disparity (Mean and Max PSDs) scores to evaluate model fairness in the context of overall model performance.

**Training Scheme**. For model training, we use a batch size of 10 for SLO fundus images and a batch size of 6 for OCT B-scans. The learning rate is set to 1e-4 with training conducted over 10 epochs and no weight decay applied. The optimization method is AdamW Loshchilov & Hutter (2019). The aforementioned hyperparameter setup is used in all experiments. For the proposed FIS, we set the fusion weight $c$ to 0.5, and $\tau = 1$. For the adversarial fair loss, we set $\lambda = 0.2$ to ensure all loss terms align on a similar scale, and we maintain the default settings for other experimental aspects. For the pretraining of FSCL, we utilize the default hyper-parameters and experimental setups provided in Wang et al. (2022). All the

Table 2: Performance on the SLO fundus images in the test set with identity **race**. FIS stands for fair identity scaling, which is the proposed method. The means and standard deviations (Means) of scores are reported based on three runs with different random seeds.

| Disease | Method | Overall AUC↑ | Asian AUC↑ | Black AUC↑ | White AUC↑ | Mean PSD↓ | Max PSD↓ | DPD↓ | DEOdds↓ |
|---|---|---|---|---|---|---|---|---|---|
| AMD | EffNet | 79.10±0.20 | 76.42±0.43 | 68.55±0.33 | 78.57±0.25 | 5.45±0.19 | 12.67±0.19 | 17.23±0.00 | 41.65±1.23 |
| | EffNet+Adv | 78.95±0.12 | 76.78±1.34 | 71.33±1.39 | 78.32±0.35 | 3.80±0.80 | 8.85±1.50 | 17.23±0.00 | **39.63±4.28** |
| | EffNet+FSCL | 79.74±0.11 | 78.12±0.04 | 71.37±2.08 | 78.77±0.15 | 4.20±1.25 | 9.28±2.77 | 17.23±0.00 | 43.40±6.54 |
| | EffNet+FIS | **79.95±0.14** | **78.78±1.76** | **73.22±0.95** | **79.18±0.21** | **3.40±0.80** | **7.46±1.86** | 17.23±0.00 | 50.16±5.49 |
| DR | EffNet | 79.25±0.07 | 67.57±0.64 | 71.88±0.94 | 81.30±0.34 | 7.24±0.55 | 17.33±1.17 | 1.78±1.54 | **13.84±3.95** |
| | EffNet+Adv | 79.43±0.24 | 66.91±0.68 | 69.92±0.69 | 81.91±0.28 | 8.16±0.26 | 18.89±1.16 | 6.51±3.36 | 26.75±9.09 |
| | EffNet+FSCL | 80.20±0.30 | 66.04±7.63 | 73.77±1.31 | 81.97±0.37 | 8.11±2.75 | 19.85±6.63 | **1.19±0.16** | 20.09±3.74 |
| | EffNet+FIS | **80.57±0.42** | **69.10±2.77** | **74.15±1.52** | **82.60±0.69** | **6.92±0.98** | **16.76±2.81** | 2.81±1.28 | 17.81±7.60 |
| Glaucoma | EffNet | 77.49±0.29 | 81.62±0.49 | 72.57±0.67 | 77.65±0.37 | 4.78±0.61 | 11.69±1.50 | 10.04±2.15 | 12.37±3.06 |
| | EffNet+Adv | 77.98±0.76 | 80.88±1.13 | 73.56±1.45 | 78.21±0.79 | 3.88±0.37 | 9.38±0.73 | 10.41±2.79 | 11.34±3.68 |
| | EffNet+FSCL | **78.31±0.10** | 81.39±0.49 | 75.27±0.41 | **78.25±0.13** | **3.20±0.12** | **7.83±0.25** | **9.53±0.49** | **10.00±2.18** |
| | EffNet+FIS | 78.27±0.43 | **81.85±0.93** | **75.61±1.33** | 78.18±0.32 | 3.27±0.78 | 7.98±1.93 | 12.15±2.25 | 8.61±2.48 |

Table 3: Performance on the OCT B-Scans in the test set with identity **race**. FIS stands for fair identity scaling, which is the proposed method.

| Disease | Method | Overall AUC↑ | Asian AUC↑ | Black AUC↑ | White AUC↑ | Mean PSD↓ | Max PSD↓ | DPD↓ | DEOdds↓ |
|---|---|---|---|---|---|---|---|---|---|
| AMD | 3D CNN | 81.24±0.48 | 80.89±0.16 | 76.29±1.71 | 80.47±0.66 | 2.56±0.91 | 5.67±2.03 | 17.23±0.00 | 37.91±6.75 |
| | 3D CNN+Adv | 81.29±0.88 | 80.00±2.36 | 72.11±3.94 | 80.88±1.08 | 4.85±1.35 | 10.78±2.74 | 17.23±0.00 | **28.01±16.71** |
| | 3D CNN+FIS | **82.62±1.40** | **82.45±4.46** | **78.69±3.44** | **81.74±1.27** | **1.97±1.20** | **4.55±2.50** | 17.23±0.00 | 34.49±4.26 |
| DR | 3D CNN | 92.22±0.53 | 95.94±3.83 | 85.13±1.63 | 93.91±0.94 | 5.09±1.74 | 11.72±3.99 | 8.89±3.21 | **6.03±8.30** |
| | 3D CNN+Adv | 92.22±0.31 | 95.13±1.73 | 87.49±1.26 | 93.26±0.59 | 3.53±0.39 | 8.29±0.87 | **0.90±0.30** | 7.19±2.46 |
| | 3D CNN+FIS | **93.27±0.13** | **96.05±1.09** | **89.09±0.75** | **94.06±0.30** | 3.14±0.44 | 7.46±1.08 | 6.35±2.29 | 17.54±5.89 |
| Glaucoma | 3D CNN | 86.49±0.19 | 88.81±0.48 | 82.90±0.52 | 86.57±0.10 | **2.81±0.11** | **6.83±0.20** | 10.20±2.87 | **4.47±6.29** |
| | 3D CNN+Adv | 86.21±0.06 | 89.37±0.33 | 81.14±1.03 | 86.70±0.17 | 3.97±0.37 | 9.54±0.82 | **3.95±8.26** | 6.29±9.79 |
| | 3D CNN+FIS | **86.96±0.04** | **90.63±0.60** | **83.23±0.31** | **86.91±0.08** | 3.47±0.30 | 8.50±0.78 | 19.73±1.36 | 18.25±5.69 |

experiments are conducted on a machine running Ubuntu 22.04, equipped with an Nvidia RTX A6000 graphics card.

## 5.2 Experimental Results

**Race Results.** The 3D OCT model (**Tables 2**) always outperforms the 2D SLO fundus model (**Table 3**) for AMD, DR and glaucoma by a margin of 2.1%, 13.0%, and 9.0% when using the baseline EfficientNet model, respectively. The performance-scaled disparity scores of Mean PSD and Max PSD of the 3D OCT model are generally about 50% lower than the 2D fundus model. For instance, the Mean PSD and Max PSD of the 3D OCT model for AMD detection are 2.56% and 5.67% compared with 5.45% and 12.67% for the 2D fundus model when using the baseline EfficientNet model. The traditional fairness metric DEO shows the same trend that 3D OCT model is more fair than 2D fundus model, while the other traditional metric DPD does not show the same finding. For the *2D fundus results*, FIS improves the overall AUC from 79.1% to 79.95%, 79.25% to 80.57%, and 77.49% to 78.27% for AMD, DR, and glaucoma, respectively. Notably, FIS improves the AUC in Blacks from 68.55% to 73.22%, 71.88% to 74.15%, and 72.57% to 75.61% for AMD, DR, and glaucoma detections, respectively, while at the same time AUCs for Whites and Asians all have 1% to 2% improvements. Consistently with the AUC improvements, Mean PSDs are reduced from 5.45% to 3.40%, 7.24% to 6.92%, and 4.78% to 3.27% and Max PSDs reduced from 12.67% to 7.46%, 17.33% to 16.76%, and 11.69% to 7.98% for AMD, DR, and glaucoma, respectively. In all three eye disease detection tasks, FIS demonstrates superior overall AUC performance and exhibits lower PSD scores compared to the two SOTA models, Adv and FSCL. For the *3D OCT results*, FIS improves the overall AUC from 81.24% to 82.62%, 92.22% to 93.27%, and 86.49% to 86.96% for AMD, DR, and glaucoma, respectively. As observed in the 2D fundus results, FIS constantly improves the AUC in Blacks from 76.29% to 78.69%, 85.13% to 89.09%, and 82.90% to 83.23% for AMD, DR, and glaucoma detections, respectively, while at the same time AUCs for Whites and Asians all have 1% to 2% improvements. FIS also reduces Mean PSD scores from 2.56% to 1.97%, 5.09% to 3.14%, and Max PSDs from 5.67% to 4.55% and 11.72% to 7.46% for AMD and DR, respectively. This inconsistent trend between the improved AUC performance and degraded PSD scores for glaucoma detection underscores the importance of using comprehensive fairness metrics for fairness assessment.

**Gender Results. Table 4** and **Table 5** show the testing performance for detecting the three eye diseases concerning the identity attribute gender using 2D SLO fundus images and 3D OCT Bscans, respectively. The 3D OCT models generally have better overall and group-

Table 4: Performance on the SLO fundus images in the test set with identity **gender**. FIS stands for fair identity scaling, which is the proposed method.

| Disease | Method | Overall AUC↑ | Female AUC↑ | Male AUC↑ | Mean PSD↓ | Max PSD↓ | DPD↓ | DEOdds↓ |
|---|---|---|---|---|---|---|---|---|
| AMD | EffNet | 79.10±0.20 | **80.58±0.30** | 76.61±0.07 | 2.51±0.15 | 5.02±0.29 | 3.74±0.00 | 8.74±5.47 |
| | EffNet+Adv | 79.25±0.41 | 80.54±0.55 | 77.06±0.20 | 2.20±0.23 | 4.39±0.45 | 3.74±0.00 | 12.18±2.48 |
| | EffNet+FSCL | 79.15±0.23 | 80.24±0.26 | 77.20±0.31 | 1.92±0.15 | 3.83±0.30 | 3.74±0.00 | **7.16±13.52** |
| | EffNet+FIS | **79.62±0.37** | 80.30±0.26 | **78.38±0.59** | **1.20±0.23** | **2.41±0.46** | 3.74±0.00 | 10.66±1.56 |
| DR | EffNet | 79.25±0.07 | 79.54±0.44 | 79.06±0.35 | 0.30±0.46 | 0.61±0.93 | 1.13±0.44 | 1.59±4.67 |
| | EffNet+Adv | 78.94±0.15 | 79.12±0.16 | 78.62±0.07 | 0.31±0.06 | 0.63±0.13 | 1.69±1.25 | 2.04±0.77 |
| | EffNet+FSCL | 79.47±0.41 | 79.07±0.46 | 79.86±0.46 | 0.50±0.10 | 0.99±0.21 | **0.04±0.82** | **0.44±0.85** |
| | EffNet+FIS | **80.43±0.76** | **80.28±0.82** | **80.48±1.07** | **0.12±0.38** | **0.24±0.75** | 0.38±0.26 | 3.41±1.55 |
| Glaucoma | EffNet | 77.49±0.29 | 76.90±0.30 | 78.17±0.30 | 0.82±0.07 | 1.64±0.14 | 4.85±0.62 | 5.22±1.44 |
| | EffNet+Adv | 77.94±0.15 | **77.42±0.32** | 78.66±0.08 | **0.79±0.25** | **1.59±0.49** | **1.22±1.58** | **0.87±1.58** |
| | EffNet+FSCL | 78.19±0.29 | 77.35±0.27 | 79.28±0.37 | 1.23±0.17 | 2.47±0.34 | 4.56±0.46 | 4.81±0.82 |
| | EffNet+FIS | **78.32±0.31** | 77.25±0.27 | **79.69±0.40** | 1.56±0.19 | 3.12±0.37 | 2.10±0.37 | 3.19±1.12 |

Table 5: Performance on the OCT B-Scans in the test set with identity **gender**. FIS stands for fair identity scaling, which is the proposed method.

| Disease | Method | Overall AUC↑ | Female AUC↑ | Male AUC↑ | Mean PSD↓ | Max PSD↓ | DPD↓ | DEOdds↓ |
|---|---|---|---|---|---|---|---|---|
| AMD | 3D CNN | 81.24±0.48 | 81.53±0.59 | 80.66±0.45 | 0.54±0.25 | 1.08±0.49 | 3.74±0.00 | 4.32±1.39 |
| | 3D CNN+Adv | 82.34±0.67 | 82.62±0.62 | 81.78±0.80 | 0.51±0.24 | 1.02±0.48 | 3.74±0.00 | **2.75±1.50** |
| | 3D CNN+FIS | **82.84±2.65** | **82.74±2.44** | **82.94±3.03** | **0.12±0.43** | **0.25±0.85** | 3.74±0.00 | 4.74±1.15 |
| DR | 3D CNN | 92.22±0.53 | 92.70±0.49 | 91.42±0.57 | 0.69±0.15 | 1.38±0.30 | 4.68±0.98 | 2.61±2.25 |
| | 3D CNN+Adv | 93.20±0.71 | 93.46±0.66 | 92.63±0.97 | **0.45±0.46** | **0.90±0.93** | 3.98±1.98 | **1.98±0.77** |
| | 3D CNN+FIS | **93.58±1.25** | **94.09±1.23** | **92.84±1.34** | 0.67±0.11 | 1.34±0.22 | 4.03±3.93 | 6.50±2.79 |
| Glaucoma | 3D CNN | 86.49±0.19 | 84.93±0.60 | 88.47±0.39 | **2.05±0.55** | **4.10±1.10** | 6.86±1.39 | 8.84±1.44 |
| | 3D CNN+Adv | 86.62±0.50 | 84.89±0.35 | 88.81±0.73 | 2.26±0.30 | 4.53±0.60 | 6.26±0.85 | **8.75±0.65** |
| | 3D CNN+FIS | **87.16±0.16** | **85.30±0.31** | **89.40±0.40** | 2.36±0.37 | 4.71±0.74 | **5.76±1.03** | 9.91±0.48 |

wise AUC performance and lower PSD scores. For the *2D fundus results*, FIS improves the overall AUC from 79.1% to 79.62%, 79.25% to 80.43%, 77.49% to 78.32% for AMD, DR, and glaucoma, respectively. FIS also reduces Mean PSD scores from 2.56% to 1.2% and 0.3% to 0.12%, and Max PSDs from 5.02% to 2.41% and 0.61% to 0.24% for AMD and DR, respectively. For the *3D OCT results*, FIS improves the overall AUC from 81.24% to 82.84%, 92.22% to 93.58%, 86.49% to 87.16% for AMD, DR, and glaucoma, respectively. In addition, FIS achieves the best AUC performance for both the female and male groups. However, our FIS model only attains best PSD scores for AMD detection.

**Ethnicity Results.** For the *2D fundus results* (**Table 6**), FIS improves the overall AUC from 79.1% to 79.53%, 79.25% to 80.81%, 77.49% to 78.55% for AMD, DR, and glaucoma, respectively. Notably, FIS improves the AUC in Hispanics from 77.77% to 84.27%, 85.37% to 85.44%, and 72.23% to 73.62% for AMD, DR, and glaucoma detections, respectively, while at the same time AUCs for Non-Hispanics all have 1% to 2% improvements. For AMD and DR detection, although our FIS model has the best overall and group-wise AUCs, the PSD scores are the worst due to the large difference between Hispanics and non-Hispanics. This contradicted phenomenon again underlines the importance of having various fairness metrics to cover all aspects of model fairness evaluation. In all three eye disease detection tasks, FIS demonstrates superior overall and group-wise AUC performance compared to the two SOTA models, Adv and FSCL, while Adv and FSCL models do not consistently transcend baseline in terms of overall and group-wise AUCs. For the *3D OCT results* (**Table 7**), FIS improves the overall AUC from 81.24% to 83.53%, 92.22% to 93.17%, and 86.49% to 86.55% for AMD, DR, and glaucoma, respectively. FIS constantly improves the AUC in non-Hispanics by about 2.3%, 1.0%, and 0.1% for AMD, DR, and glaucoma detections, respectively, while the AUCs for Hispanics have about 1% drop for AMD detection and 0.2% and 0.5% increase for DR and glaucoma detection, respectively. FIS also reduces Mean PSD scores from 4.79% to 2.41%, 0.57% to 0.11%, and 2.33% to 2.05%, and Max PSDs from 9.58% to 4.81%, 1.14% to 0.22%, and 4.66% to 4.09% for AMD, DR, and glaucoma, respectively. Again, unlike the Mean and Max PSD scores, DPD and DEOdds results are less consistent with overall and group-wise AUCs results.

**Effects of Fusion Weight** $c$. The hyperparameter $c$ plays a central role in our proposed FIS. To gain insights into its impact on fairness learning, we conducted an ablation analysis with identity race, visualized in **Figure 2**. The results reveal a notable trend: FIS,

Table 6: Performance on the SLO fundus images in the test set with identity **ethnicity**. FIS stands for fair identity scaling, which is the proposed method.

| Disease | Method | Overall AUC↑ | Non-Hisp AUC↑ | Hispanic AUC↑ | Mean PSD↓ | Max PSD↓ | DPD↓ | DEOdds↓ |
|---------|--------|--------------|---------------|---------------|-----------|----------|------|---------|
| AMD | EffNet | 79.10±0.20 | 79.16±0.22 | 77.77±2.25 | 0.88±0.98 | 1.75±1.95 | 9.31±0.00 | 23.46±4.00 |
| | EffNet+Adv | 78.99±0.10 | 79.05±0.18 | 77.63±2.83 | 0.90±1.22 | 1.79±2.43 | 9.31±0.00 | 21.63±1.94 |
| | EffNet+FSCL | 79.36±0.23 | 79.39±0.20 | 78.24±1.39 | **0.72±0.53** | **1.45±1.05** | 9.31±0.00 | 21.95±4.11 |
| | EffNet+FIS | **79.53±0.10** | **79.42±0.04** | **84.27±2.38** | 3.04±1.47 | 6.09±2.95 | 9.31±0.00 | **19.08±2.46** |
| DR | EffNet | 79.25±0.07 | 78.84±0.09 | 85.37±1.65 | 4.12±1.04 | 8.24±2.09 | 4.55±0.71 | 24.31±4.27 |
| | EffNet+Adv | 79.25±0.32 | 79.04±0.31 | 80.75±2.67 | **1.08±1.79** | **2.16±3.58** | **0.54±4.25** | 5.93±13.35 |
| | EffNet+FSCL | 79.84±0.60 | 79.64±0.61 | 81.56±2.18 | 1.20±1.36 | 2.41±2.72 | 0.94±4.38 | **4.39±13.09** |
| | EffNet+FIS | **80.81±0.81** | **80.46±0.77** | **85.44±2.73** | 3.08±1.49 | 6.17±2.96 | 7.87±4.10 | 16.99±9.80 |
| Glaucoma | EffNet | 77.49±0.29 | 77.69±0.28 | 72.23±1.30 | 3.53±0.77 | 7.05±1.55 | 1.31±1.24 | 6.28±1.94 |
| | EffNet+Adv | 77.67±0.21 | 77.93±0.30 | 70.87±2.59 | 4.55±1.86 | 9.09±3.71 | 6.22±2.00 | 14.44±6.14 |
| | EffNet+FSCL | 78.02±0.60 | 78.25±0.61 | 72.30±0.39 | 3.82±0.51 | 7.63±1.03 | 1.71±1.27 | **5.67±1.87** |
| | EffNet+FIS | **78.55±0.37** | **78.74±0.43** | **73.62±1.75** | **3.26±1.34** | **6.52±2.68** | **1.25±1.40** | 8.67±4.50 |

Table 7: Performance on the OCT B-Scans in the test set with identity **ethnicity**. FIS stands for fair identity scaling, which is the proposed method.

| Disease | Method | Overall AUC↑ | Non-Hisp AUC↑ | Hispanic AUC↑ | Mean PSD↓ | Max PSD↓ | DPD↓ | DEOdds↓ |
|---------|--------|--------------|---------------|---------------|-----------|----------|------|---------|
| AMD | 3D CNN | 81.24±0.48 | 81.01±0.45 | **88.80±2.22** | 4.79±1.13 | 9.58±2.26 | 9.31±0.00 | **11.15±4.20** |
| | 3D CNN+Adv | 82.95±0.62 | 82.83±0.57 | 86.96±3.21 | 2.49±1.40 | 4.97±2.80 | 9.31±0.00 | 18.74±8.36 |
| | 3D CNN+FIS | **83.53±1.76** | **83.41±1.83** | 87.43±1.71 | **2.41±1.88** | **4.81±3.77** | 9.31±0.00 | 16.93±3.33 |
| DR | 3D CNN | 92.22±0.53 | 92.08±0.17 | 93.13±2.90 | 0.57±1.22 | 1.14±2.45 | 5.01±2.27 | 10.24±3.83 |
| | 3D CNN+Adv | 92.93±1.06 | 92.96±1.15 | 92.18±0.22 | 0.42±0.46 | 0.84±0.90 | 5.72±3.14 | **3.17±4.80** |
| | 3D CNN+FIS | **93.17±0.59** | **93.13±0.56** | **93.33±1.37** | **0.11±0.37** | **0.22±0.74** | **4.10±1.14** | 8.46±2.18 |
| Glaucoma | 3D CNN | 86.49±0.19 | 86.67±0.21 | 82.64±0.65 | 2.33±0.49 | 4.66±0.98 | 7.90±3.46 | 15.87±4.99 |
| | 3D CNN+Adv | 86.02±0.12 | 86.20±0.11 | 81.78±0.76 | 2.57±0.38 | 5.13±0.76 | **3.42±1.48** | **8.91±1.81** |
| | 3D CNN+FIS | **86.55±0.52** | **86.68±0.50** | **83.14±1.43** | **2.05±0.58** | **4.09±1.16** | 7.10±1.97 | 16.85±3.45 |

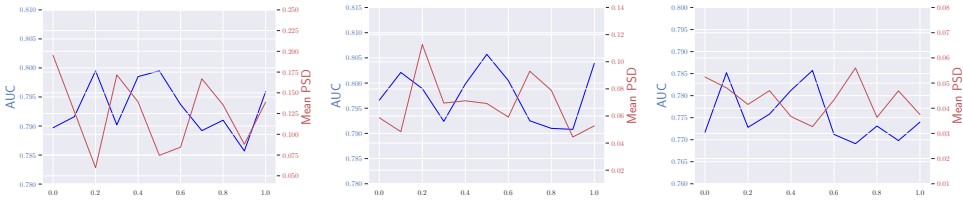

Figure 2: Effects of the fusion weight $c$ on AUC and Mean PSD in AMD detection (left), DR detection (middle), and glaucoma detection (right). SLO fundus images are used for this analysis.

which combines both group-level and individual-level information, consistently outperforms scenarios involving only individual-level scaling ($c = 0$) or group-level scaling ($c = 1$). When $c$ is set to 0.5, we strike a balance between performance and demographic equity across all three disease types, demonstrating the effectiveness of this parameter in achieving a desirable trade-off.

## 6 CONCLUSIONS

While minority groups experience more health issues, there are currently no large-scale medical datasets with 3D imaging data and comprehensive demographic identity attributes available for thorough fairness learning. In this paper, we present our Eye Fairness datasets with 30,000 subjects with 2D fundus images and 3D optical coherence tomography scans covering three major eye diseases causing irreversible vision loss including age-related macular degeneration, diabetic retinopathy, and glaucoma affecting 380 million patients worldwide. Along this large-scale eye disease screening dataset, we propose a fair identity scaling method combining group and individual scaling to improve model fairness, which demonstrates superior performance compared with various SOTA models. To facilitate fair comparisons between different fairness learning models, we propose a new fairness metric named performance-scaled disparity that is intuitive to be understood by medical practitioners. We expect that our Eye Fairness dataset can significantly contribute to new fairness learning model development in the machine learning and computer vision research community.

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

## A   Appendix

### A.1   Computational Cost for Training

Training an EfficientNet for 10 epochs in the AMD detection task requires 66 milliseconds per SLO fundus image and 833 milliseconds for an OCT B-scan. On the other hand, when training an EfficientNet with the proposed FIS for the same task, it takes 71 milliseconds per SLO fundus image and 901 milliseconds for an OCT B-scan. This indicates a slight increase in training time with the FIS method compared to the baseline EfficientNet, due to the additional fairness-related computations involved in FIS.

### A.2   Effects of Various Backbone Models

**Figure 3** presents a performance comparison among various backbone models for AMD detection using both SLO fundus images and OCT B-scans in relation to racial identity. EfficientNet demonstrates superior performance in SLO fundus images, a 2D imaging modality. In contrast, since OCT B-scans represent a 3D imaging modality, one might consider using a CNN with 3D convolutional modules. As indicated in **Figure 3**, a straightforward 3D CNN surpasses other models, underscoring the compatibility between the 3D imaging modality and 3D convolutional operations.

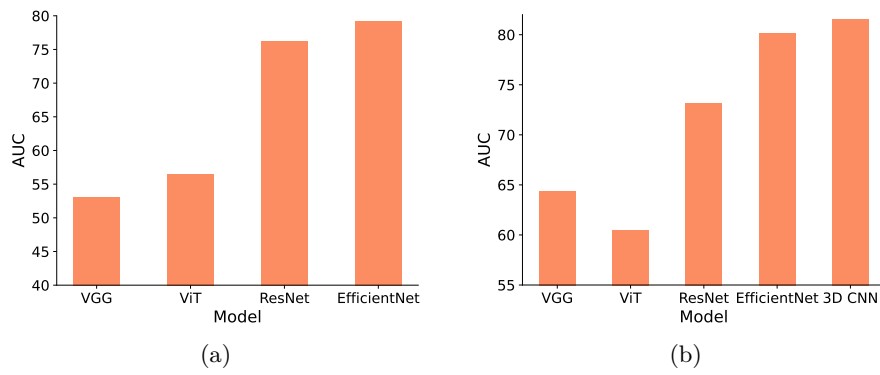

Figure 3: Comparison of performance across different backbone models for AMD detection using SLO fundus images (a) and OCT B-scans (b) w.r.t. the racial identity.

