# OpenReview forum: "Eye Fairness: A Large-Scale 3D Imaging Dataset for Equitable Eye Diseases Screening and Fair Identity Scaling"
_ICLR.cc/2024/Conference — Submitted to ICLR 2024_

### Official Review · Reviewer_8dk2 · 2023-10-12

**Soundness:** 2 fair
**Presentation:** 2 fair
**Contribution:** 3 good
**Rating:** 5
**Confidence:** 5

**Summary:**

The paper introduces the EyeFairness dataset, aimed to promote the fairness study for medical imaging. The dataset comprises 30,000 subjects with both 2D and 3D imaging data, capturing various demographic attributes. Additionally, the authors propose a fair identity scaling (FIS) approach to enhance model fairness for this dataset.

**Strengths:**

- This paper studies an important topic of fairness for medical imaging. The authors introduce a relatively large-scale dataset for 2D fundus photos and 3D OCT scans. It covers major eye diseases and captures a few different demographic attributes, which can be a useful resource for the community.
- The authors propose Fair Identity Scaling (FIS) to improve the fairness of the model.

**Weaknesses:**

- The authors didn't tune the hyper-parameters of the baseline methods but only used the default HPs, which leads to unfair comparisons. The baseline methods are not designed for medical imaging, so if applied to a different setting, the hyperparameters should be carefully tuned to get the best performance. Especially there're adversarial training method and self-supervised pretraining method that are very sensitive to the HPs.
- The definition of performance-scaled disparity (PSD) is not clear. It says in the paper: "PSD metrics are calculated as the standard deviation of group performance or absolute maximum group performance difference divided by overall performance." Which one did the authors use? And what does Mean PSD and Max PSD mean?
- Also, regarding the metrics, authors can provide the worst-case AUC and the AUC gap between best-performing and worst-performing groups besides the current overall AUC and group-wise AUC. It would be clearer to directly look at the AUC gap to validate the effectiveness of the proposed methods. Additionally, what are the advantages of using PSD instead of the AUC gap? Consider an extreme case: for model 1, two groups have an AUC of 25% and 51% while for model 2, two groups have an AUC of 50% and 100%. According to the authors' definition, $PSD_1 = (51-25)/51=0.51 < PSD_2 = (100-50)/100=0.5$, but can you say model 2 is fairer than model 1 as it's the smaller the better? I know the AUC usually is higher than 50% but this is just an example and I think there are many similar cases in regular scenarios.
- I think Table 1 in this paper is taken from Table 1 in [1] without reference as (1) the number of images of each dataset is not the original number but the number after preprocessing by [1] and (2) the so-called ADNI 1.5T is a subset of the large ADNI dataset [2] extracted by [1].
- At the time I wrote this review, the GitHub repo authors provided was empty.
- Minor: the current citation style makes reading difficult. The author should use (Deng et al. (2009)) instead of Deng et al. (2009), i.e. \citep instead of \cite.

[1] Zong et al. MEDFAIR: Benchmarking Fairness for Medical Imaging. ICLR'23.\
[2] Wyman et al. Standardization of analysis sets for reporting results from adni mri data. Alzheimer’s & Dementia 2013.

**Questions:**

- Can the authors also provide further breakdown statistics of the intersectional groups, e.g. black females?
- The dataset also contains some other attributes such as preferred language. I'm not very sure how this is related to eye diseases and fairness, e.g. do non-English speaking patients get lower AUC? But how does the eye imaging model perceive the speaking? Also, the authors did not evaluate the performance of different subgroups of preferred language and marital status.

---

> ### Author Response · Authors · 2023-11-22
> **Response (Part 1 out of 2)**
>
> **Question**: The authors didn't tune the hyper-parameters of the baseline methods but only used the default HPs, which leads to unfair comparisons. The baseline methods are not designed for medical imaging, so if applied to a different setting, the hyperparameters should be carefully tuned to get the best performance. Especially there're adversarial training method and self-supervised pretraining method that are very sensitive to the HPs.
>
> **Response**: We have meticulously tuned the hyperparameters for all baseline models presented in this paper. It's important to note that without such tuning, the baseline approaches yield inferior results. To enhance clarity and provide a comprehensive understanding, we have included a detailed discussion on this aspect in the revised manuscript.
>
>
> **Question**: The definition of performance-scaled disparity (PSD) is not clear. It says in the paper: "PSD metrics are calculated as the standard deviation of group performance or absolute maximum group performance difference divided by overall performance." Which one did the authors use? And what does Mean PSD and Max PSD mean?
>
> **Response**: PSD, performance-scaled disparity, aims to use a single scaler to quantify group disparities based on the AUC metric. The PSD formula is defined as follows:
> $$
> \text{Mean PSD} = \frac{ \text{STD}(\text{Group-wise AUCs}) } {\text{Overall AUC}}
> $$
> $$
> \text{Max PSD} = \frac{ \text{Max}(\text{Group-wise AUCs}) -  \text{Min}(\text{Group-wise AUCs})  } {\text{Overall AUC}}
> $$
>
> We have revised the manuscript accordingly to improve the clarity.
>
> **Question**: Also, regarding the metrics, authors can provide the worst-case AUC and the AUC gap between best-performing and worst-performing groups besides the current overall AUC and group-wise AUC. It would be clearer to directly look at the AUC gap to validate the effectiveness of the proposed methods. Additionally, what are the advantages of using PSD instead of the AUC gap? Consider an extreme case: for model 1, two groups have an AUC of 25% and 51% while for model 2, two groups have an AUC of 50% and 100%. According to the authors' definition,, but can you say model 2 is fairer than model 1 as it's the smaller the better? I know the AUC usually is higher than 50% but this is just an example and I think there are many similar cases in regular scenarios.
>
>
> **Response**: We thank the reviewer for the comment. The table of the performance on Race for AMD detection is as follows. FIS continues to demonstrate superior performance in fairness metrics. Furthermore, as outlined in the definition of Max PSD,  Max PSD is closely correlated with AUC gap that is part of Max PSD. We fully agree that the PSD metric has its own limitations in representing the model fairness in scenarios outlined by the reviewer. The advantage of PSD is that: when the AUC gap is the same for two models, PSD considers the model with a higher AUC performance as the fairer model, while the AUC gap itself in this case considers the two models are equally fair. Realizing the fact that each fairness metric may have its limitations, we also calculated other fairness metrics in our manuscript such as DPD and DEOdds.
>
>
> | Method | Worst-case AUC | AUC Gap |
> |----------|----------|----------|
> | EffNet | 68.55 |  10.02 |
> | EffNet+Adv | 71.33 | 7.62 |
> | EffNet+FSCL | 71.37 | 7.40 |
> | EffNet+FIS | 73.22 |  5.96 |
>
>
> **Question**: I think Table 1 in this paper is taken from Table 1 in [1] without reference as (1) the number of images of each dataset is not the original number but the number after preprocessing by [1] and (2) the so-called ADNI 1.5T is a subset of the large ADNI dataset [2] extracted by [1].
>
> **Response**: We thank the reviewer for pointing this out. We have cited the paper of MedFair [1] and clarified that ADNI 1.5T is a subset of the large ADNI dataset extracted by [1] in the revision.
>
>
> **Question**: At the time I wrote this review, the GitHub repo authors provided was empty.
>
> **Response**: We apologize for the delay in making our dataset and code available. They were released for peer review on October 16th at https://github.com/anonymous4science/EyeFairness. To comply with IRB requirements, we meticulously reviewed our dataset including 30,000 patients to ensure the removal of any identifiable information, such as names, locations, and dates, before the data release.
>
> **Question**: Minor: the current citation style makes reading difficult. The author should use (Deng et al. (2009)) instead of Deng et al. (2009), i.e. \citep instead of \cite.
>
> **Response**: We have fixed our citation style in the revised manuscript.

---

> ### Author Response · Authors · 2023-11-22
> **Response (Part 2 out of 2)**
>
> **Question**: Can the authors also provide further breakdown statistics of the intersectional groups, e.g. black females?
>
> **Response**: The statistics of the intersectional groups (race + gender) are presented in the following table. We have added more statistics of intersectional groups to the revised manuscript.
>
> | Combination | AMD | DR | Glaucoma |
> |----------|----------|----------|----------|
> | Asian Female | 6.78 | 8.07 | 8.41 |
> | Asian Male | 4.93   | 6.50 | 6.46 |
> | Black Female | 3.89 | 4.11 | 4.49 |
> | Black Male | 3.17   | 3.53 | 3.98 |
> | White Female | 48.00 | 43.35 | 44.13 |
> | White Male  | 33.23 | 34.44 | 32.53 |
>
>
> **Question**: The dataset also contains some other attributes such as preferred language. I'm not very sure how this is related to eye diseases and fairness, e.g. do non-English speaking patients get lower AUC? But how does the eye imaging model perceive the speaking? Also, the authors did not evaluate the performance of different subgroups of preferred language and marital status.
>
> **Response**: Our dataset is from the United States, where a patient's primary language can be potentially indicative of their socioeconomic status. This factor could impact the fairness of AI models. For instance, patients with lower incomes, who might seek hospital care later and often present with multiple confounding eye/systemic diseases, could influence the predictions of deep learning models. The research [2] has shown that non-English speakers often have diseases at more advanced stages when they seek medical help.
>
> Following the suggestion, the experimental results of preferred language and marital status are reported in the first and second tables below, respectively. FIS demonstrates consistent superiority over the baseline model, showcasing enhanced generalizability across metrics such as AUC, Mean PSD, and Worst-case AUC. Notably, the AUC gap when compared to EfficientNet is narrower than that with EfficientNet+FIS. This indicates that FIS not only enhances the worst-case AUC but also improves the best-case AUC. The results, along with a relevant discussion, have been included in the revised manuscript.
>
> | Method | AUC | Mean PSD | Worst-case AUC | AUC Gap |
> |----------|----------|----------|----------|----------|
> | EffNet | 78.99 |  6.76 | 73.70 |  11.72 |
> | EffNet+FIS | 80.17 |  6.24 | 74.87 |  12.24 |
>
> | Method | AUC | Mean PSD | Worst-case AUC | AUC Gap |
> |----------|----------|----------|----------|----------|
> | EffNet | 78.99 |  1.19 | 76.97 |  2.13 |
> | EffNet+FIS | 79.65 |  0.83 | 78.51 |  1.74 |
>
>
> **References**:
>
> [1] Zong, Yongshuo, Yongxin Yang, and Timothy Hospedales. "MEDFAIR: Benchmarking Fairness for Medical Imaging." The Eleventh International Conference on Learning Representations. 2022.
>
> [2] Halawa, Omar A., et al. "Race and ethnicity differences in disease severity and visual field progression among glaucoma patients." American journal of ophthalmology 242 (2022): 69-76.

---

### Official Review · Reviewer_7iiL · 2023-10-29

**Soundness:** 3 good
**Presentation:** 3 good
**Contribution:** 4 excellent
**Rating:** 6
**Confidence:** 4

**Summary:**

The paper presents a new large-scale dataset for eye disease diagnostic comprising of 30’000 2D fundus as well as 3D OCT images for AMD retinopathy and glaucoma diagnostics. In addition to some baseline comparison on fairness metics it also proposes a new fair identity scaling.

**Strengths:**

The proposed dataset is very valuable as it is very large in size (30’000 patients with 2D and 3D imaging) covering three relevant eye diseases. The analysis is strong and the explored fair identity scaling is a reasonable approach to address inequality in datasets in general. Providing (to my understanding) paired 2D fundus and 3D OCT imaging could also pave the way for new hybrid diagnostic tools.

**Weaknesses:**

Overall, despite its value, the proposed dataset is somewhat limited in that it seems to be acquired from a single centre in the US. A pooling with previous public datasets would likely increase its value and reduce the “unfairness” by design (rather than re-weighting).
The statement “effective image augmentation strategies for 3D imaging data are largely unclear” is wrong in my opinion and there is no citation that backs it up. Many 3D medical image analysis methods make good use of image augmentation strategies.
The chosen baselines are rather simple and no true SOTA results are presented. The presentation of the results is mainly focussed on numerical comparison and I would be missing a more in-depth analysis or discussion why certain races perform better or worse. E.g. since Hispanics are under-represented in the dataset it is not intuitive that this group achieves the highest AUC without and not with FIS.
It remains unclear whether the data is always “paired”, ie. the same patient is measured with 2D and 3D imaging.

**Questions:**

Clarify the 2D/3D data split wrt. patients. Discuss a pooling of other public datasets from centres with different scanners / from different countries. Expand the baselines and correct the statement of non-existing 3D augmentation strategies.

---

> ### Author Response · Authors · 2023-11-22
> **Response**
>
> **Question**: Discuss a pooling of other public datasets from centres with different scanners / from different countries
>
> **Response**: As shown in Table 1 in the manuscript, for eye diseases, there are no public fairness datasets available that have comprehensive demographic identity attributes as in our dataset including age, gender, race, ethnicity, preferred language, and marital status. Our dataset is also the only public dataset with 3D eye imaging data measured by optical coherence tomography that enables 3D fairness learning modeling. For instance, the ODIR-2019 dataset only contains demographic identity attributes of age and gender, which is insufficient for comprehensive fairness learning. Another example is that the AMD-OCT dataset only contains the age attributes. Therefore, given the uniqueness of our dataset, which covers three major eye diseases, including age-related macular degeneration, diabetic retinopathy, and glaucoma, affecting 380 million patients worldwide, contains a large number of patients of 30,000 in total and six demographic attributes (age, gender, race, ethnicity, preferred language, and marital status) for fairness learning, there are no existing public datasets that can be used to pool with our dataset. In addition, we would like to stress that our dataset is from a prominent eye hospital located in a region with a large population with high diversities.
>
>
> | Dataset       | Imaging Modality | Number of Patients  | Disease Focus | Identity Attribute |
> |----------|----------|----------|----------|----------|
> | ODIR-2019      | Fundus          | 5,000  | AMD, Glaucoma, DR | Age; Gender  |
> | AMD-OCT        | OCT                |  384   | AMD |   Age   |
>
>
>
> **Question**: Clarify the 2D/3D data split wrt. patients.
>
> **Response**: As detailed in Section 5.1, our dataset utilizes a patient-level split. It comprises one 2D fundus image and a set of 3D OCT B-Scans for each patient. This information has been further clarified in the revised version of our manuscript.
>
> **Question**: Expand the baselines.
>
> **Response**: We are grateful for the reviewer's insightful comment. We have included the state-of-the-art fairness method, FairAdaBN [1], in our experiments. The results for AMD detection based on race are presented below. We tune the hyperparameters of FairAdaBN to ensure optimal performance and utilize its best-reported backbone network, ResNet. However, it's important to note that FairAdaBN's dependency on the backbone network architecture restricts its generalizability and ability to leverage more powerful pre-trained models, such as EfficientNet. We have updated the experiment in the revised manuscript accordingly.
>
> | Method | AUC | Mean PSD |
> |----------|----------|----------|
> | FairAdaBN [1] | 78.88 |  3.63 |
> | EffNet | 79.10 |  5.45 |
> | EffNet+Adv | 78.95 | 3.80 |
> | EffNet+FSCL | 79.74 | 4.20 |
> | EffNet+FIS | 79.95 |  3.40 |
>
> **Question**: Correct the statement of non-existing 3D augmentation strategies.
>
> **Response**: We appreciate the reviewer’s contribution to this discussion. Rather than delving into general 3D augmentation strategies, our discussion focus is on augmentation methods specifically pertinent to fairness. To our knowledge, there is a lack of established fairness methods demonstrating that algorithmic fairness can be improved through 3D augmentation. The application of current 3D augmentation techniques, such as cutmix, random crop, and mixup, raises concerns. These methods could inadvertently lead to inaccurate disease detection by obscuring or eliminating abnormal lesions in the 3D space. This discussion has been included in the revised manuscript.
>
> **References**:
>
> [1] Zikang Xu, Shang Zhao, Quan Quan, Qingsong Yao, and S. Kevin Zhou. 2023. FairAdaBN: Mitigating Unfairness with Adaptive Batch Normalization and Its Application to Dermatological Disease Classification. In Medical Image Computing and Computer Assisted Intervention – MICCAI 2023: 26th International Conference, Vancouver, BC, Canada, October 8–12, 2023, Proceedings, Part II. Springer-Verlag, Berlin, Heidelberg, 307–317.

---

### Official Review · Reviewer_fenP · 2023-10-31

**Soundness:** 3 good
**Presentation:** 4 excellent
**Contribution:** 3 good
**Rating:** 5
**Confidence:** 4

**Summary:**

This paper introduces the EyeFairness dataset that includes both 2D fundus photos and 3D optical coherence tomography (OCT) scans, together with six demographic features (age, gender, race, ethnicity, preferred language, marital status), and proposes a fair identity scaling (FIS) approach to combine group and individual scaling to improve model fairness. FIS was demonstrated to improve performance in eye disease screening according to fairness metrics, when implemented together with EfficientNet-B1, against other fairness methods. Fairness methods are especially appropriate in the eye screening domain due to known differing burdens of eye diseases amongst ethnicities.

**Strengths:**

-	FIS exploits both group and individual scaling to manage within-group sample variation
-	Detailed comparison of proposed FIS against other fairness methods (Adv, FSCL) on various demographic features

**Weaknesses:**

-	Minimal ablation analysis temperature scaling parameter, FIS group/individual scaling trade-offs actually not very consistent (Figure 2)
-	Side-effects of fairness on adjacent demographic features not considered

**Questions:**

1. The main contribution of a Fair Identity Scaling (FIS) model with learnable group weights and past individual loss data, might have been analyzed in greater details as to the temperature scaling parameter (alongside fusion weight).
2. Conceptually, the distinction between improvements in “general (AUC) performance” and “fairness” might have been further considered. In particular, from the results in Tables 2 to 7, FIS appears often capable of not only improving “fairness” (i.e. minimizing performance-scaled disparity), but instead often improves performance in all groups (and overall). As such, a natural question might be whether FIS might be used with arbitrary groupings of data, to improve classifier performance.
3. Returning to fairness as a focus, the presented analysis does not appear to be concerned with the impact on fairness amongst other demographic features, when FIS is applied to a particular feature. For example, when FIS is applied on race (as in Tables 2 & 3), what is the effect on results stratified with other features such as gender, ethnicity, age etc.? This appears particularly relevant since the other demographics may be no less significant for the consideration of fairness/equity purposes.
4. The costs of considering fairness might be discussed in greater detail, in particular the possibility that optimizing the proposed PSD metric possibly reduces overall classification performance (and thus medical care). This is because PSD (and other fairness metrics) emphasize between-group equality, which may come at the cost of reduced aggregate performance (although this is largely not the case in this study).

---

> ### Author Response · Authors · 2023-11-22
> **Response (Part 1 out of 2)**
>
> **Question**: Minimal ablation analysis temperature scaling parameter, FIS group/individual scaling trade-offs actually not very consistent (Figure 2) / The main contribution of a Fair Identity Scaling (FIS) model with learnable group weights and past individual loss data, might have been analyzed in greater details as to the temperature scaling parameter (alongside fusion weight).
>
> **Response**: As suggested by the reviewer,  we have included a table below showcasing the experimental results for AMD detection with different temperature values, specifically focusing on the attribute of race in fundus images, with the hyperparameter c set to 0.5. This analysis has been incorporated into the revised version of our paper.
>
> | $\tau$ | AUC | Mean PSD |
> |----------|----------|----------|
> |0.2 | 77.25 |  0.0565 |
> |0.4 | 77.86 | 3.68 |
> |0.8 | 79.39 | 5.59 |
> |1 | 79.95 | 3.40 |
> |2 | 79.54 | 5.18 |
> |3 | 79.49 | 5.01 |
> |4 | 79.60 | 7.02 |
> |5 | 79.49 | 5.07 |
>
> **Question**: Side-effects of fairness on adjacent demographic features not considered/Returning to fairness as a focus, the presented analysis does not appear to be concerned with the impact on fairness amongst other demographic features, when FIS is applied to a particular feature. For example, when FIS is applied on race (as in Tables 2 & 3), what is the effect on results stratified with other features such as gender, ethnicity, age etc.? This appears particularly relevant since the other demographics may be no less significant for the consideration of fairness/equity purposes.
>
> **Response**: We thank the reviewer for the insightful comment. Focusing on a single attribute can make the cause-and-effect relationship clear. However, as the reviewer highlighted, examining the impact of applying FIS to one attribute on the performance metrics of other attributes is also valuable. To this end, we have conducted an analysis where FIS is specifically applied to race in the context of AMD detection, and we report its effects on gender and ethnicity performance. For our FIS model specifically addressing the fairness for race, the model fairness quantified by PSD values for gender and ethnicity attributes is also improved. It's important to note that the individual scaling in Equation (1) is attribute-independent and is designed to adaptively influence the significance of each sample during back-propagation. This approach could be a contributing factor to the observed improvements in fairness.
>
> | Method | AUC (Race) | Mean PSD (Race) | Mean PSD (Gender) | Mean PSD (Ethnicity) |
> |----------|----------|----------|----------|----------|
> | EffNet | 79.10 |  5.45 |  7.24 |  4.78 |
> | EffNet+FIS (Race) | 79.95 |  3.40 |  1.58 |  0.58 |
>
>
>
> **Question**: Conceptually, the distinction between improvements in “general (AUC) performance” and “fairness” might have been further considered. In particular, from the results in Tables 2 to 7, FIS appears often capable of not only improving “fairness” (i.e. minimizing performance-scaled disparity), but instead often improves performance in all groups (and overall). As such, a natural question might be whether FIS might be used with arbitrary groupings of data, to improve classifier performance.
>
> **Response**: The observed improvement in both fairness and performance can be attributed to the individual scaling mechanism in FIS, which considers the losses of samples within a mini-batch to determine their relative importance for back-propagation. This approach enables the training process to give priority to samples with larger losses, effectively balancing the learning focus across the dataset.
>
> Following the suggestion, we present the experimental results for arbitrary groupings in AMD detection, as detailed in the table below. We observe that combining race, gender, and ethnicity enhances performance, although the resulting mean PSDs are not consistently optimal compared to other methods. This highlights the intricate and intriguing interplay between attributes. We plan to delve deeper into this aspect in our future research.
>
>
> | Method | AUC (Race) | Mean PSD (Race) | Mean PSD (Gender) | Mean PSD (Ethnicity) |
> |----------|----------|----------|----------|----------|
> | EffNet | 79.10 |  5.45 |  7.24 |  4.78 |
> | EffNet+FIS (Race) | 79.95 |  3.40 |  1.58 |  0.58 |
> | EffNet+FIS (Race+Gender) | 79.84 | 3.90 | 1.40 | 0.38 |
> | EffNet+FIS (Race+Gender+Ethnicity) | 80.28 | 3.95 | 1.30 | 0.32 |

---

> > ### Author Response · Authors · 2023-11-22
> > **Response (Part 2 out of 2)**
> >
> > **Question**: The costs of considering fairness might be discussed in greater detail, in particular the possibility that optimizing the proposed PSD metric possibly reduces overall classification performance (and thus medical care). This is because PSD (and other fairness metrics) emphasize between-group equality, which may come at the cost of reduced aggregate performance (although this is largely not the case in this study).
> >
> > **Response**: PSD, performance-scaled disparity, is an evaluation metric that uses a single scaler to quantify group disparities based on the AUC metric. The PSD formula is defined as follows:
> > $$
> > \text{Mean PSD} = \frac{ \text{STD}(\text{Group-wise AUCs}) } {\text{Overall AUC}}
> > $$
> > $$
> > \text{Max PSD} = \frac{ \text{Max}(\text{Group-wise AUCs}) -  \text{Min}(\text{Group-wise AUCs})  } {\text{Overall AUC}}
> > $$
> >
> > It's important to note that PSD is exclusively an evaluation metric and does not influence the training process. This approach is consistent with standard evaluation practices in research involving DPD and DEodds, as exemplified in papers like D. Zietlow et al., 2022. We have clarified this aspect further in our revised manuscript.

---

### Official Review · Reviewer_eXPH · 2023-11-06

**Soundness:** 3 good
**Presentation:** 2 fair
**Contribution:** 4 excellent
**Rating:** 6
**Confidence:** 3

**Summary:**

This paper proposes a publicly available large-scale (30,000 subjects) 3D eye imaging dataset (OCT/Fundus) for disease screening and fair identity scaling. The authors also propose a fair identity scaling metric to evaluate model performance.

**Strengths:**

1. Addressing the fairness issue is an important topic and organizing such a large-scale dataset including three types of measurements: 1) retinal imaging 2) demographic group information 3) disease diagnosis requires a large amount of effort.

2. The authors ran several baselines (EfficientNet, 3D CNN) and evaluated the classification with some fairness metrics (e.g. PSD, DPD).

**Weaknesses:**

1. The abstract and introduction section is quite lengthy. The core contributions and the highlights can be combined.

2. Some of the writing needs to be improved (e.g. "model performance across different models").

**Questions:**

1. The trend of hyperparameter $c$ in Figure 2 is not quite clear since it kind of alternates for both AUC and mean PSD. The authors might need to further discuss the choice of $c$, which still seems rather empirical given the current visualization.

2. The experimental results section is currently flooded with numbers and quite hard to follow.

---

> ### Author Response · Authors · 2023-11-22
> **Response**
>
> **Question**: The trend of hyperparameter  in Figure 2 is not quite clear since it kind of alternates for both AUC and mean PSD. The authors might need to further discuss the choice of, which still seems rather empirical given the current visualization.
>
> **Response**: We adhere to the Keep It Simple and Stupid (KISS) principle by setting the default hyperparameter c to 0.5, striking a balance between individual and group scaling to achieve a trade-off between effectiveness/accuracy and fairness. $c = 1$ represents exclusive group scaling, while $c = 0$ denotes solely individual scaling. As corroborated by the analysis in Figure 2, our choice of c at 0.5 aligns with empirical results, demonstrating that higher AUC coincides with lower mean PSD, compared to other c values.
>
>
> **Question**: The experimental results section is currently flooded with numbers and quite hard to follow.
>
> **Response**: We appreciate the reviewers' feedback on the comprehensiveness of our study, which encompasses a wide range of factors such as identity attributes, metrics, and methods, and includes extensive experiments across three eye diseases. Recognizing the importance of thoroughness in our experimental results, we have enhanced the readability of our experimental section by incorporating visualizations in the revised manuscript.

---

### Author Response · Authors · 2023-11-22
**Summary of Responses**

Dear AC and Reviewers,

We sincerely thank AC and all reviewers for their time and efforts in reviewing our manuscript. Their constructive feedback has significantly contributed to the enhancement of our paper.

We are thankful for the reviewers' recognition of the significance of our introduced dataset in fairness research. As highlighted in the introduction, age-related macular degeneration (AMD), diabetic retinopathy (DR), and glaucoma collectively impact a vast population globally, with 200 million, 103 million, and 80 million patients affected, respectively. Due to fellow eye compensation and brain interpolation, a substantial number of patients are not aware that they have eye diseases and therefore are left undiagnosed without timely treatment. For instance, it has been reported that half of all glaucoma patients are undiagnosed and Blacks are 4 times more likely to have undiagnosed glaucoma than Whites. Deep learning screening using retinal imaging data has proven an affordable way for eye disease screening to reduce health disparity, especially in minorities and socioeconomically disadvantaged groups. Deep-learning systems for screening eye diseases must be proven to be fair in order to maximize their benefits. The imperative of fairness in medical tasks extends beyond ethical considerations, encompassing legal and medical obligations to ensure equitable healthcare for all. As shown in Table 1, for eye diseases, there are no public fairness datasets available that have comprehensive demographic identity attributes as in our dataset including age, gender, race, ethnicity, preferred language, and marital status. Our dataset is also the only public dataset with 3D eye imaging data measured by optical coherence tomography that enables 3D fairness learning modeling. Therefore, given the uniqueness of our dataset, which covers three major eye diseases, including age-related macular degeneration, diabetic retinopathy, and glaucoma, affecting 380 million patients worldwide, contains a large number of patients of 30,000 in total and six demographic attributes (age, gender, race, ethnicity, preferred language, and marital status) for fairness learning, the release of this dataset is poised to greatly advance the understanding and application of fairness in AI and computer vision research.

In addition to our dataset contribution, our fair identity scaling is a flexible module that can be conveniently used with the existing backbone models to promote model performance equity, which has been shown in our comprehensive experiments.

We believe the combination of our major contribution of releasing the large-scaling fairness learning dataset for eye disease screening and our minor contribution of developing the fair identity scaling model constitutes a significant positive impact on the AI and computer vision research community.

We made our dataset and code publicly available for peer review on October 16th at https://github.com/anonymous4science/EyeFairness. It's important to note the extensive manual effort involved in ensuring compliance with IRB requirements. This process included meticulously checking our dataset including 30,000 patients for any identifiable information (such as names, locations, dates, etc.) before releasing the data.

In response to your comments and suggestions, we have made several significant updates to our manuscript, which we summarize as follows:

1. We have included an ablation study examining the effects of temperature alongside fusion weight.

2. An analysis has been added to explore the side-effects of fairness on adjacent demographic features.

3. We have incorporated a state-of-the-art fairness method into our experiments for comparative analysis.

4. To enhance our understanding of fairness assessment, we now report scores for both worst-case AUC and AUC gap.

5. Detailed statistics for intersectional groups have been provided for a more comprehensive view.

6. We present experimental results focusing on the attributes of preferred language and marital status.

Our responses aim to resolve the reviewers' concerns and improve the clarity of the paper. We are happy to answer any additional questions and provide more information.



Sincerely,

Authors

---

### Meta-Review · Area_Chair_whTX · 2023-12-11

**Metareview:**

This paper introduces the EyeFairness dataset, comprising 30,000 subjects with 2D and 3D imaging data, is recognized as a significant resource for eye disease diagnostics and fairness studies in medical imaging. The paper is commended for its focus on fairness in medical imaging and the introduction of the fair identity scaling (FIS) approach. The dataset's extensive coverage of demographic attributes and the proposed FIS method's ability to manage within-group sample variation are acknowledged as notable strengths. Additionally, the detailed comparison of FIS against other fairness methods, particularly its exploitation of both group and individual scaling, is highlighted as a positive aspect, contributing to the understanding of fairness considerations in medical imaging applications. The weaknesses identified in the four reviews converge on several key points. Firstly, concerns are raised about the paper's lack of comprehensive consideration for computational costs or scalability issues associated with the proposed fair identity scaling (FIS) method, with specific emphasis on the absence of a code link. Reviewers also question the clarity and completeness of the presentation of results, noting that the experimental section is challenging to follow due to the abundance of numerical information. Additionally, there are requests for a more in-depth analysis and discussion regarding the performance disparities among different demographic groups, particularly when FIS is applied. Several reviewers express a need for expanded baselines, more extensive hyperparameter tuning, and additional fairness metrics for a thorough evaluation. The absence of a clear distinction between improvements in general performance and fairness, as well as potential side-effects of fairness on adjacent demographic features, is underscored as a point of concern. Furthermore, reviewers recommend addressing issues related to the unclear definition of performance-scaled disparity (PSD) metrics and the potential limitations of PSD compared to other fairness metrics. The rebuttal has made efforts to address a majority of the concerns raised. However, it has not garnered strong support from the reviewers post-rebuttal, primarily stemming from persistent issues related to the clarity and unpreparedness of certain sections of the manuscript. Furthermore, there remains ambiguity about how the paper can provide benefits to the broader audience within ICLR who may not specialize in medical image analysis. Considering the overall assessment of the paper, the reviews, and the rebuttal, the meta-reviewer recommends rejection. The meta-reviewer, however, does recommend the authors consider resubmitting the manuscript to medical journals or conferences once it has undergone further preparation and improvement based on the reviews.

**Justification For Why Not Higher Score:**

Reviewers raise concerns about result presentation clarity, excessive numerical information, and insufficient analysis of performance disparities. They recommend improvements in baselines, fairness metrics, and hyperparameter tuning. Issues with performance-scaled disparity (PSD) metrics and manuscript clarity persist post-rebuttal. The paper's relevance to a broader ICLR audience remains unclear. The meta-reviewer recommends rejection but suggests resubmitting to medical journals or conferences after further preparation.

**Justification For Why Not Lower Score:**

N/A

---

### Decision · Program_Chairs · 2024-01-16

Reject